# Screening for and Disclosure of Domestic Violence during the COVID-19 Pandemic: Results of the PRICOV-19 Cross-Sectional Study in 33 Countries

**DOI:** 10.3390/ijerph20043519

**Published:** 2023-02-16

**Authors:** Elizaveta Fomenko, Ines Keygnaert, Esther Van Poel, Claire Collins, Raquel Gómez Bravo, Päivi Korhonen, Merja K. Laine, Liubove Murauskiene, Athina Tatsioni, Sara Willems

**Affiliations:** 1Department of Public Health and Primary Care, Ghent University, 9000 Ghent, Belgium; 2Research Centre, Irish College of General Practitioners, D02 XR68 Dublin, Ireland; 3Centre Hospitalier Neuropsychiatrique, Rehaklinik, L-9002 Ettelbruck, Luxembourg; 4Research Group Self-Regulation and Health, Institute for Health and Behaviour, Department of Behavioural and Cognitive Sciences, Faculty of Humanities, Education and Social Sciences, University of Luxembourg, L-4366 Esch-sur-Alzette, Luxembourg; 5Department of General Practice, University of Turku and Turku University Hospital, 20521 Turku, Finland; 6Department of General Practice and Primary Health Care, University of Helsinki and Helsinki University Hospital, 00014 Helsinki, Finland; 7Folkhälsan Research Center, 00280 Helsinki, Finland; 8Public Health Department, Faculty of Medicine, Vilnius University, LT-01513 Vilnius, Lithuania; 9Research Unit for General Medicine and Primary Health Care, Faculty of Medicine, School of Health Sciences, University of Ioannina, 45110 Ioannina, Greece

**Keywords:** domestic violence, screening, disclosure, general practitioner, primary health care, COVID-19, PRICOV-19, infectious diseases, quality of care, equity

## Abstract

The COVID-19 pandemic left no one untouched, and reports of domestic violence (DV) increased during the crisis. DV victims rarely seek professional help, yet when they do so, they often disclose it to their general practitioner (GP), with whom they have a trusting relationship. GPs rarely screen and hence rarely take the initiative to discuss DV with patients, although victims indicate that offering this opportunity would facilitate their disclosure. This paper aims to describe the frequency of screening for DV by GPs and disclosure of DV by patients to the GP during the COVID-19 pandemic, and to identify key elements that could potentially explain differences in screening for and disclosure of DV. The PRICOV-19 data of 4295 GP practices from 33 countries were included in the analyses, with practices nested in countries. Two stepwise forward clustered ordinal logistic regressions were performed. Only 11% of the GPs reported (much) more disclosure of DV by patients during COVID-19, and 12% reported having screened for DV (much). Most significant associations with screening for and disclosure of DV concerned general (pro)active communication. However, (pro)active communication was performed less frequently for DV than for health conditions, which might indicate that GPs are insufficiently aware of the general magnitude of DV and its impact on patients and society, and its approach/management. Thus, professional education and training for GPs about DV seems highly and urgently needed.

## 1. Introduction

In March 2020, policymakers worldwide took measures to slow down and limit the spread of SARS-CoV-2, including lockdowns, home-isolation, and the restriction of free movement. These measures taken to fight the spread of the virus had a significant impact at individual, social, and economic levels. A forced stay at home, combined with the uncertainty of the pandemic and its progression, unstable socio-economic conditions, financial constraints, and physical threats, among others, increased the tension at home for many families, fuelling feelings of stress, anxiety, and frustration. Moreover, isolation increases the exposure to exploitative relationships, psychological stress, and lack of support or limited access to the protective network, which, combined with the pandemic measures, transform homes into a dangerous place for (potential) victims of domestic violence (DV). The escalation of tension and conflicts in the household might trigger or worsen different forms of DV [1]. As happened in previous crises such as natural disasters and wars, the COVID-19 pandemic has also seen an increase in reported DV cases [2,3,4,5,6,7,8,9,10,11,12]. 

DV is a global public health problem of pandemic proportions. It comprises any behaviour in the family or domestic context causing physical, psychological, sexual or socio-economic suffering to someone else [13]. Irrespective of whether the victim(s) and assailant(s) share biological or legal family ties, the assailant(s) and victim(s) may (have) live(d) at the same residence. DV can thus also occur between both current and former (intimate) partners [14], being the most frequent form of DV, also called intimate partner violence (IPV), affecting one in three women worldwide. However, DV goes beyond IPV and also includes child abuse and neglect, violence between siblings or relatives, and elder abuse [15,16,17].

Considering DV from a public health perspective, DV not only has important physical, sexual, reproductive, psychological, neurobiological, social, and economic implications for victims, but also for their families, communities, and even future generations [7,18,19,20,21,22,23,24,25,26,27]. However, research has shown that victims of DV rarely seek professional/formal help specifically for those who have experienced DV for some time [23,28,29], as they find it hard to disclose the experience of DV by themselves [30,31,32,33]. Nevertheless, when they seek help upon victimization, they often contact their general practitioner (GP) [28,29,31]. While GPs rarely ask about experiences of DV [30,32], primary care is considered as the most appropriate setting for talking about DV, and victims indicate that GPs should actively screen for DV to facilitate disclosure [29,32]. GPs, and by extension all primary care professionals, are in a privileged position to facilitate disclosure in a safe environment [17]. By routinely asking, GPs can provide primary, secondary or tertiary prevention, increasing early detection and preventing the recurrence of events or progression by following up the victims and their families and offering proper referrals to the right services if needed [16,34]. Routine screening habits can be considered as a form of desirable prevention, in contrast to general prevention, which may be thought of as initiatives which anticipate problems in a planned and methodical approach. Aiming to improve or safeguard the health and wellbeing of the target group, desirable prevention is defined as “initiatives which anticipate problems even earlier in a targeted and systematic way, are maximally “of-fensive,” use an integral approach, work in a participatory manner, and have a democratic character” [35,36]. Additionally, multidisciplinary collaboration is key to procuring any kind of help and support needed and addressing the consequences of DV. Failure to receive appropriate help in time can aggravate the underlying problem and further increase the risk of serious, long-term, and sometimes life-threatening complaints in victims, families, communities, and future generations [37,38,39].

The aim of this paper is (a) to describe the frequency of screening for DV by GPs and disclosure of DV by patients to the GP during the COVID-19 pandemic, and (b) to identify key elements that could potentially explain differences in screening for and disclosure of DV by using a rather phenomenological and exploratory approach, in line with desirable prevention. 

## 2. Materials and Methods

### 2.1. Study Design and Setting

In the summer of 2020, an international consortium of more than 45 research institutes was formed under the coordination of Ghent University (Belgium) to set up the study to consider how primary care practices were organized during the COVID-19 pandemic (PRICOV-19). This multi-country cross-sectional study focused on quality and safety in primary care practices during the COVID-19 pandemic. Data were collected in 38 countries by means of an online self-reported validated questionnaire among primary care practices. The questionnaire was developed at Ghent University in multiple phases, including a pilot study among 159 general practices in Flanders (Belgium). More details are described elsewhere [40,41]. The questionnaire was translated into 38 languages following a standard procedure. The Research Electronic Data Capture (REDCap) platform was used to host the questionnaire in all languages, send out invitations to the national samples of general/family practices, and securely store the answers from the participants [42].

### 2.2. Sampling and Recruitment

The data reported here were collected between November 2020 and December 2021, except for Belgium, where data were partially collected from 20 May 2020 onwards. The timeframe for data collection varied between countries from 3 to 35 weeks. In each partner country, the consortium partner(s) recruited primary care practices following a predefined recruitment procedure [40], which is shown in the Appendix A (Appendix A). Sampling and recruitment are detailed elsewhere [40,43]. One questionnaire was completed per practice, at least one reminder was sent in all countries, and the overall response rate was 27.8%. 

### 2.3. Measurements

#### 2.3.1. Characteristics of the Respondent and the Practice

This paper only used the questionnaires completed by GPs and GP trainees. Questionnaires completed by other health professionals working in a GP practice were not included. In all instances in the analysis where we refer to GP, this includes GPs and GP trainees unless otherwise stated. The questionnaire contained questions regarding the respondent (position in practice and years of work experience) and the practice (number of GPs and GP trainees working in practice and location of the practice). Additionally, respondents were also asked to compare their practice’s average patient population (including patients with a migration background with difficulty mastering the local language, with limited or low health literacy, with financial problems, psychiatric vulnerability, over the age of 70, chronic conditions, little social support or limited informal care) to other practices in their country through seven 3-point Likert-scale items, ranging from ‘Below average (1)’ to ‘Above average (3)’.

#### 2.3.2. Patient Flow

Patient flow related questions included questions on the appointment system (walk-in hours and video consultations), triage (protocol when answering phone calls, telephonic triage, available information on how to refer a patient to a triage station), and changes in the GP’s role (involvement in actively reaching out to patients that might postpone healthcare, increased responsibilities, task shifting and further training). Additionally, respondents were asked about the occurrence of specific incidents since the COVID-19 pandemic due to the complexity of primary care and the high degree of uncertainty (e.g., ‘A patient with a serious condition was seen late because he/she did not know how to call on a GP’). These five coded as ‘yes (1)’ or ‘no (0)’ items were summed into one scale ranging from 0 to 5, Cronbach’s alpha = 0.752. 

#### 2.3.3. (Pro-)Active Communication

Specific initiatives to contact vulnerable patients (with a chronic disorder, a chronic condition that needed follow-up care, psychologically vulnerable patients and/or those with previous problems of domestic violence or with a problematic child-rearing situation) were reported through four items coded as ‘yes (1)’ or ‘no (0)’. GPs were asked about their screening habits for domestic violence: ‘To what extent have you checked with patients to determine if they (in)directly experienced domestic violence since the COVID-19 pandemic?’; and financial problems: ‘To what extent have you checked with patients to determine if they experienced financial problems since the COVID-19 pandemic?’; and the occurrence of disclosure of domestic violence: ‘To what extent have patients talked to you about domestic violence since the COVID-19 pandemic?’; and financial problems: ‘To what extent have patients talked to you about financial problems since the COVID-19 pandemic?’ through four 5-point Likert-scale items, ranging from ‘Not at all (0)’ to ‘Much more than before (5)’. These items were rescaled to 3-point Likert-scale items ‘Not at all or less than before (0), ‘As much as before (1)’ and ‘(Much) more than before (2)’ as there were insufficient units of observation in the extremities. 

Finally, the questionnaire also contained four ‘yes (1)’/‘no (0)’ items concerning communication (practice website, leaflet, answering machine, and COVID-19 specific leaflet) with patients. These items were summed into one scale ranging from 0 to 4, Cronbach’s alpha = 0.654.

#### 2.3.4. Wellbeing of the Respondent

The wellbeing of the GPs was assessed using the expanded 9-item Mayo Clinic Wellbeing Index (eWBI) [41]. Seven items are responded to with a ‘yes (1)’ or ‘no (0)’, and the remaining two items are responded to on a 7-point or 5-point Likert scale, ranging from ‘strongly disagree’ to ‘strongly agree’. The first seven items are summed into one scale ranging from 0 to 7, Cronbach’s alpha = 0.791. If the respondents replied ‘strongly disagree’ or ‘disagree’ one point was added to their score, those who reported ‘agree’ or ‘strongly agree’ had one point subtracted from their score. No adjustments were made for those with middle/neutral responses. Being at risk of distress is defined as a score of ≥2, as per previous studies [41].

### 2.4. Data Analysis

Data was imported into SPSS27 for initial data cleaning and manipulation. All statistical analyses were conducted with R software version 4.0.3 on the database of 33 countries, available as of 3 November 2021. (a) To describe the sample, as well as the frequency of screening for DV by GPs and disclosure of DV by patients to the GP during the COVID-19 pandemic, simple descriptive statistics were analysed, and group differences were computed using a (post hoc) chi-square test as all assumptions of the chi-square test could be met. (b) To identify key elements that could potentially explain differences in screening for and disclosure of DV (between those who answered ‘Not at all or less than before’, ‘As much as before’ and ‘(Much) more than before’ to the screening for and disclosure of DV questions) we chose an exploratory study. Two stepwise forward-clustered ordinal logistic regressions were therefore conducted to analyse the association between the multiple factors and the two outcome variables. To avoid multicollinearity, the correlations were checked between all variables. There were no strong correlations (<0.60). To ascertain the power required for the multiple logistic regression analyses, we assumed a ratio of ten cases per predictor based on the simulation by Peduzzi et al. [44]. The respondents were clustered by country, leading to a multilevel analysis. Main terms with *p* < 0.05 were included in the multivariate models, and variables that produced at least one beta estimate significantly different from zero were retained. It was also determined whether these added main effects significantly improved the prediction of both outcome variables using a likelihood ratio test. Furthermore, the Akaike information criterion (AIC) and Bayesian information criterion (BIC) were used to compare the relative quality of one model to another by balancing a model’s goodness of fit against its complexity. By doing so, it takes into account the risk of overfitting as well as the risk of underfitting. Models were then ranked from best to worst, with the “best” model showing the smallest AIC and BIC. Ordinal logistic regression was chosen because the outcome variables consisted of three categories (not at all/less than before = 0, as much as before = 1, (much) more than before = 2). Logit is the most commonly used link, and it allows the production of odds ratios by exponentiation of the model estimates. Negative estimates of the independent variables show that one value of an independent variable compared to its following value is more likely to receive lower values on the ordinal dependent variable and vice versa for positive estimates.

### 2.5. Ethical Approval

The study was conducted according to the guidelines of the Declaration of Helsinki. The Research Ethics Committee of Ghent University Hospital approved the protocol of the PRICOV-19 study (BC-07617). Research ethics committees in the different partner countries gave additional approval if needed in that country. All participants gave informed consent on the first page of the online questionnaire.

## 3. Results

The analysis included 4295 GPs with a valid value for the variables concerning DV. A description of the main characteristics of the sample is shown in Table 1. Approximately one-quarter of the respondents were in each 10-year age group. In terms of location, the majority (43.2%) of the respondents were working in practices based in cities/suburbs and 34.6% work in single-handed practices. Screening for and disclosure of DV during the COVID-19 pandemic was significantly more likely to have been undertaken by respondents working in practices that estimated the proportion of patients with limited health literacy, financial problems, psychiatric vulnerability, and/or little social support in their practice above average for their country. Additionally, disclosure of DV by the patient was significantly more likely reported by respondents working in practices that considered their proportion of patients with a migration background above average. 

Compared to before the pandemic, almost half of the respondents (47.9%) reported no disclosure or less disclosure by the patient, four out of ten (40.9%) reported as much disclosure as before and only 11.4% reported (much) more disclosure during the pandemic. These proportions varied per country, as shown in Figure 1, but there are, however, almost no countries reporting much more disclosure during compared to before the pandemic. 

Similar results were found for the screening for DV by the GP, with four out of ten respondents (38.7%) who did not screen or screened less for DV during compared to before the pandemic. Almost half of the respondents (49.2%) screened as much as before, and only 12.1% screened (much) more than before the pandemic (Figure 2).

Appendix A Appendix A (Disclosure of DV by the patient) and Appendix A (Screening for DV by the GP) present the results of the multilevel ordinal logistic regression analysis. Model I, showing the intercept-only model, has an ICC of, respectively, 10.5% and 9.9%, meaning that approximately 10% of the variance in both, disclosure of and screening for DV, is attributable to the country. Each subsequent stepwise model shows a better goodness-of-fit (based on a smaller AIC and/or BIC value). Variances at the group and individual levels reduce when adding variables, indicating that the large variance in disclosure of and screening for DV between countries is reduced by adding individual-level variables.

Reporting a perceived higher or equal average number of patients with a psychiatric vulnerability in practice compared to the rest of the country, (much) more screening for or disclosure of financial problems during the COVID-19 pandemic, and actively reaching out to patients with a known history of DV or problematic child-rearing situations during the COVID-19 pandemic were all positively associated with (much) more disclosure of and screening for DV. On the other hand, a greater number of GPs in practice and a prevalence of five different types of incidents (where patients were seen late) were only associated with (much) more disclosure of DV by the patient, while the presence of print versions of the triage information in every consultation room, GPs who (strongly) agreed that they were more involved in actively reaching out to patients who might postpone healthcare during the COVID-19 pandemic, and the presence of at least three communication items (e.g., leaflets, website, answering machine) for the patients in the GP’s practice, were associated with (much) more screening for DV by the GP (Table 2).

## 4. Discussion

Since the start of the COVID-19 pandemic, experts feared an increase in DV due to the isolation, movement-restricting measures, uncertainty, and impact of the pandemic on the socio-economic conditions, health threats, etc. As the crisis progressed, reports of DV did increase [6,7,8,9,10]. Hence, an expected need for the most accessible caregivers, namely GPs during the pandemic, to address this risk in their patients. Yet, in the 33 countries participating in this study, only 12% of the GPs reported having screened more or much more for DV than before the pandemic, and 11% of the GPs reported more or much more frequent disclosure of DV by the patients. On the other hand, half of the GPs reported no disclosure at all or less than before the pandemic, and four out of ten GPs did not screen or screened less than before the pandemic. This confirms previous research [29,45,46,47] that found that most GPs are insufficiently aware of the general magnitude of DV and its impact on people directly and indirectly exposed to it, or inadequately trained to approach and manage it.

Based on our study, screening for and disclosure of DV during the COVID-19 pandemic occurred significantly more often during compared to before the pandemic in practices that estimated the proportion of patients with limited health literacy, financial problems, a psychiatric vulnerability, and/or little social support in their practice above average for their country. Additionally, disclosure of DV by the patient also occurred significantly more often during compared to before the pandemic in practices that considered their proportion of patients with a migration background above average. Awareness of GPs, concerning risk factors of DV, is a major step to identify vulnerable people. However, these results could not be entirely confirmed with the regression analyses. Only a higher estimated proportion of patients with a psychiatric vulnerability in practice came out as a significant predictor of screening for, as well as disclosure of DV during the COVID-19 pandemic. This suggests that GPs who estimated a higher proportion of vulnerable patients within their practice, compared to the average in the country, might not necessarily have a higher proportion of vulnerable patients, but are rather more aware of the risk of DV that these groups of vulnerable patients might face. However, while it is important to have increased attention to certain vulnerable groups, it is also important to keep in mind that anyone can become a victim of DV, especially in stressful times such as a COVID-19 pandemic with isolation and movement-restricting measures. To achieve this awareness and knowledge, appropriate training to implement tools and integrated models of care are needed. However, despite the acknowledgement of the need for further professional education/training for GPs about DV, Silva et al. found that this does not occur in the majority of courses [48]. Specializing young GPs may include practical training periods among disadvantaged patient groups/living areas or patients with a migration background. In addition, community-oriented approaches that include pathways or interventions, which may provide appropriate care to DV victims after they are identified should also be in place. The availability of integrated intervention increased preparedness after training in managing the complexity of DV in clinical practice; however, it was insufficient to catalyse identification and specialist referral. The lack of guidance in the usage of information about DV also has been reported as a barrier in practice [49]. 

According to our study, most significant associations concern (pro)active communication. More disclosure of and screening for DV was associated with GPs who reported (much) more disclosure of and/or screening for patients’ financial problems, and who have actively been reaching out to patients with a history of DV or problematic child-rearing situations during the pandemic. Additionally, GPs who reported to have been more involved in actively reaching out to patients, in general, during the pandemic, and those who had at least three communication items (e.g., leaflets, website, answering machine) for the patients in the GP’s practice, were also more likely to screen for DV. (Pro)active communication that has been culturally adapted for the population seems to enhance trust and openness towards the patients and may help them disclose sensitive situations such as DV during consultations. However, victims generally find it difficult to raise the topic of DV themselves, and it has been evidenced that many of them prefer that the GP starts the conversation and asks directly about it. GPs, however, tend to wait for a specific clue or for the patient to address a possible history of DV, instead of routinely raising the topic without any suspicion or specific reason [30,31,32,33]. The researchers argue that the strong commitment of key individuals to addressing DV is vital for response sustainability, and would be bolstered by prioritizing DV response as a national health priority with dedicated resourcing [50]. Towards that direction, there is adequate evidence that available screening instruments can identify DV [51]. These instruments need to be translated and culturally adapted in different countries so that GPs can be trained in their use and interpretation. Previous studies supported that effective interventions generally included ongoing support services that focused on counselling and home visits, addressed multiple risk factors (not just IPV), or included parenting support for new mothers [51]. GPs need to be aware of the supportive services that are available in their community and refer victims who disclose DV to these services. The agreement on and dissemination of guidelines on screening for potential exposure to DV, the identification of barriers and facilitators for implementing such screening, and the training of GPs in interviewing victims of DV, providing these victims first with psychological help and a warm referral when needed, needs to be prioritized in the health policy agenda.

Interestingly no clear evidence was found for practice-related factors. More or much more disclosure of DV by the patient to the GP was associated with practices with more than one GP. This association was, however, not be found for screening for DV. No significant associations were found in the task-shifting variables due to the COVID-19 pandemic or the mental wellbeing of the GPs. While evidence has accumulated of increased workload, overburden, and burnout of primary care physicians during the COVID-19 pandemic [43,52], it did not seem to affect screening habits of the GP, or stop/encourage patients to disclose during the COVID-19 pandemic. 

### Strengths and Limitations

The study was conducted in 33 different countries. Although the response rate was only 28%, the analysis included 4295 GPs with a valid value for the variables concerning DV. Respondent GPs represented well each 10-year age groups of GPs currently working in primary health care. Most GP practices were based in cities or suburbs, but rural GP practices were quite well represented.

The countrywide spread of the COVID-19 pandemic and announced lockdowns varied greatly between the European countries during data collection. Given the voluntary nature of the participation of the GPs, a selection bias is possible. The study questions asked about DV compared with the situation before COVID-19 pandemic, so the GPs in the different groups (much more, as much as before, and not at all/less) may be very different from each other. The study is a cross-sectional study, thus causal relationships could not be established. Another limitation may be related to the question that was asked concerning DV. We asked about screening/disclosure compared to the time period before the COVID-19 pandemic. When GPs responded “much more than before” we do not know if they already screened a lot and now even more, or if they never screened before while now they have started screening the population. Furthermore, when they say “less than before”, it might be that they screened absolutely everyone in practice, but due to the COVID-19 pandemic they have less time and now “only” screened 95% of the patients. We do not know all of this for sure, so the GPs in each response category group (much more, as much as before, and not at all/less) can be very different.

## 5. Conclusions

While reports of DV increased during the COVID-19 pandemic, only a fraction of the GPs in our study reported (much) more screening for or disclosure of DV. According to our results, most of the significant associations with screening for and disclosure of DV concerned (pro)active communication, such as screening for other issues such as financial problems, or actively reaching out to vulnerable patients with, for example, a history of DV or problematic child-rearing situations. This leads us to believe that GPs are first of all insufficiently aware of the general magnitude of DV and its impact in society, as confirmed in previous research. Further professional education/training for GPs about DV is therefore highly and urgently needed. Such training should be implemented as a basic academic course in the GP curriculum. Additionally, community-oriented approaches that include pathways or interventions should also be in place. Secondly, once the GPs are sufficiently aware of the supportive services that are available in their community, as well as sufficiently trained (e.g., in the identification of barriers and facilitators, in interviewing skills, in offering first psychological help, etc.) and own enough tools to help them support and refer victims of DV, guidelines on screening should be agreed. This should go hand in hand with the appropriate resourcing of PC to undertake this screening and ongoing support.

## Figures and Tables

**Figure 1 ijerph-20-03519-f001:**
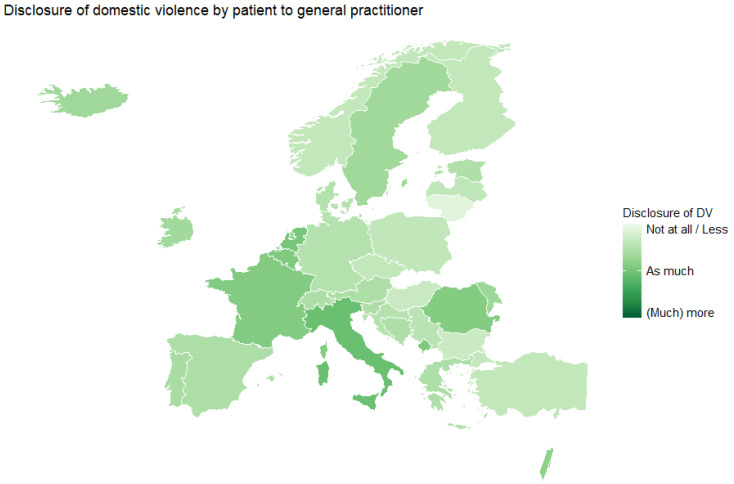
Disclosure of DV by the patient to the GP or GP trainee compared to before the COVID-19 pandemic, average score by country.

**Figure 2 ijerph-20-03519-f002:**
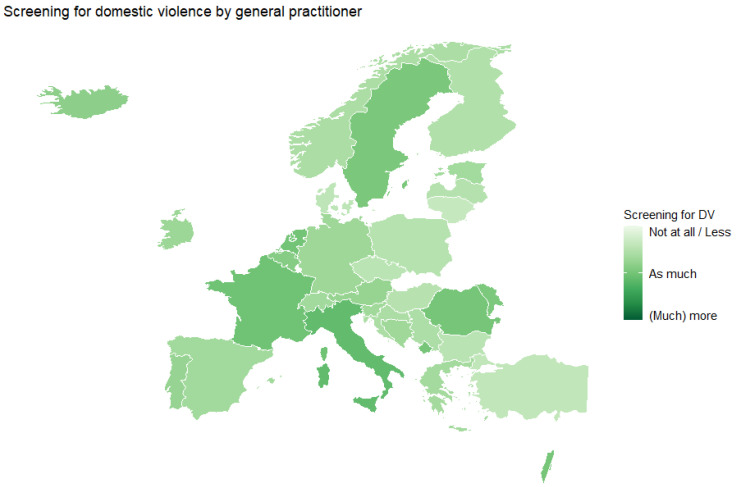
Screening for DV by the GP or GP trainee during the COVID-19 pandemic, average score by country.

**Table 1 ijerph-20-03519-t001:** Main characteristics of the general practitioners and their practices during the COVID-19 pandemic (*n* = 4295).

	Total	Disclosure of DV				Screening for DV			
	n (%)	% Not at All/Less	% as Much	% (Much) More	χ^2^; df; *p*-Value	% Not at All/Less	% as Much	% (Much) More	χ^2^; df; *p*-Value
**GP Individual Factors**									
Position					5.58; 2; 0.062				8.41; 2; 0.015
GP	4022 (93.6)	48.0 ^a^	41.0 ^a^	11.1 ^a^		38.8 ^a^	49.5 ^a^	11.8 ^a^	
GP trainee	273 (6.4)	45.4 ^a^	38.8 ^a^	15.8 ^b^		37.7 ^a^	44.7 ^a^	17.6 ^b^	
Years of experience					27.49; 8; <0.001				28.00; 8; <0.001
0–9	1075 (25.0)	44.4 ^a^	43.2 ^a^	12.5 ^a,b^		38.5 ^a^	49.7 ^a^	11.8 ^a^	
10–19	1012 (23. 6)	46.7 ^a,b^	41.7 ^a^	11.6 ^b^		36.4 ^a^	50.9 ^a^	12.7 ^a^	
20–29	1117 (26.0)	51.5 ^b^	39.1 ^a^	9.4 ^b^		41.5 ^a^	47.6 ^a,b^	10.9 ^a^	
30–39	883 (20.6)	49.4 ^a,b^	40.2 ^a^	10.4 ^b^		37.9 ^a^	51.0 ^a^	11.1 ^a^	
Unknown	208 (4.8)	44.2 ^a,b^	36.5 ^a^	19.2 ^a^		39.4 ^a^	38.9 ^b^	21.6 ^b^	
**Practice Factors**									
Location of practice					15.19; 10; 0.125				16.47; 10; 0.087
Big (inner) city	1422 (33.1)	50.4 ^a^	37.6 ^a^	12.0 ^a^		40.6 ^a^	47.2 ^a^	12.2 ^a^	
Suburbs	435 (10.1)	43.2 ^a^	43.7 ^a^	13.1 ^a^		37.2 ^a^	50.8 ^a^	12.0 ^a^	
(Small town)	776 (18.1)	49.0 ^a^	40.5 ^a^	10.6 ^a^		42.4 ^a^	47.0 ^a^	10.6 ^a^	
Mixed urban-rural	866 (20.2)	45.2 ^a^	43.8 ^a^	11.1 ^a^		35.9 ^a^	50.8 ^a^	13.3 ^a^	
Rural	783 (18.2)	47.4 ^a^	42.3 ^a^	10.3 ^a^		35.2 ^a^	52.4 ^a^	12.4 ^a^	
Unknown	13 (0.3)	46.2 ^a^	38.5 ^a^	15.4 ^a^		53.8 ^a^	38.5 ^a^	7.7 ^a^	
Number of GPs					59.40, 8, <0.001				36.09; 8; <0.001
1	1487 (34.6)	54.3 ^a^	36.2 ^a^	9.4 ^a^		42.8 ^a^	46.8 ^a^	10.4 ^a^	
2	644 (15.0)	42.7 ^b,c^	43.8 ^b^	13.5 ^b,c^		36.8 ^a,b^	49.4 ^a^	13.8 ^a,b^	
3–4	877 (20.4)	40.5 ^c^	44.9 ^b^	14.6 ^c^		33.3 ^b^	51.3 ^a^	15.4 ^b^	
5+	1231 (28.7)	47.3 ^b^	42.2 ^b^	10.5 ^a,b^		37.9 ^a,b^	50.9 ^a^	11.1 ^a^	
Unknown	56 (1.3)	58.9 ^a,b,c^	33.9 ^a,b^	7.1 ^a,b,c^		53.6 ^a^	37.5 ^a^	8.9 ^a,b^	
Patients with migration background with difficulty speaking the local language in the practice					62.58; 6; <0.001				53.04; 6; <0.001
Below average	2237 (52.1)	50.6 ^a^	39.1 ^a,b^	10.3 ^a^		41.1 ^a^	47.5 ^a^	11.4 ^a^	
Approx. average	1094 (25.5)	44.8 ^b^	44.4 ^c^	10.8 ^a^		34.4 ^b^	53.8 ^b^	11.8 ^a,b^	
Above average	733 (17.1)	39.4 ^b^	43.7 ^b,c^	16.9 ^b^		33.0 ^b^	51.4 ^a,b^	15.6 ^b^	
Unknown	231 (5.4)	61.0 ^c^	32.0 ^a^	6.9 ^a^		54.1 ^c^	35.9 ^c^	10.0 ^a,b^	
Patients with limited health literacy or low literacy in the practice					88.56; 6; <0.001				60.75; 6; <0.001
Below average	1474 (34.3)	52.7 ^a^	38.6 ^a,b^	8.7 ^a^		41.9 ^a^	48.2 ^a^	9.8 ^a^	
Approx. average	1862 (43.3)	48.6 ^b^	42.7 ^b^	10.8 ^a^		37.6 ^a^	50.1 ^a^	12.3 ^a^	
Above average	772 (18.0)	38.2 ^c^	43.3 ^b^	18.5 ^b^		31.2 ^b^	51.7 ^a^	17.1 ^b^	
Unknown	187 (4.4)	62.6 ^a^	29.4 ^a^	8.0 ^a^		55.1 ^c^	36.9 ^b^	8.0 ^a^	
Patients with financial problems in the practice					95.30; 6; <0.001				54.46; 6; <0.001
Below average	923 (21.5)	52.9 ^a,b^	38.0 ^a^	9.1 ^a^		42.0 ^a^	47.2 ^a,b^	10.7 ^a,b^	
Approx. average	2284 (53.2)	48.5 ^b^	41.9 ^a^	9.5 ^a^		38.2 ^a^	50.3 ^b^	11.5 ^b^	
Above average	919 (21.4)	38.1 ^c^	43.5 ^a^	18.4 ^b^		33.2 ^b^	50.3 ^b^	16.5 ^c^	
Unknown	169 (3.9)	63.3 ^a^	26.6 ^b^	10.1 ^a^		56.8 ^c^	38.5 ^a^	4.7 ^a^	
Patients with psychiatric vulnerability in the practice					121.88; 6; <0.001				76.33; 6; <0.001
Below average	683 (15.9)	59.6 ^a^	32.5 ^a^	7.9 ^a^		49.5 ^a^	39.7 ^a^	10.8 ^a^	
Approx. average	2646 (61.6)	47.6 ^b^	42.3 ^b^	10.1 ^a^		37.4 ^b^	51.1 ^b^	11.5 ^a^	
Above average	794 (18.5)	35.6 ^c^	45.8 ^b^	18.5 ^b^		31.0 ^c^	52.6 ^b^	16.4 ^b^	
Unknown	172 (4.0)	60.5 ^a^	27.9 ^a^	11.6 ^a,b^		51.2 ^a^	41.9 ^a,b^	7.0 ^a^	
Patients over the age of 70 in the practice					8.80; 6; 0.185				10.46; 6; 0.106
Below average	525 (12.2)	47.2 ^a^	39.8 ^a^	13.0 ^a^		34.7 ^a^	52.4 ^a^	13.0 ^a^	
Approx. average	2019 (47.0)	48.2 ^a^	40.3 ^a^	11.5 ^a^		39.7 ^a,b^	48.5 ^a^	11.7 ^a^	
Above average	1673 (39.0)	47.0 ^a^	42.4 ^a^	10.6 ^a^		38.1 ^a,b^	49.3 ^a^	12.6 ^a^	
Unknown	78 (1.8)	57.7 ^a^	28.2 ^a^	14.1 ^a^		51.3 ^b^	41.0 ^a^	7.7 ^a^	
Patients with chronic diseases in the practice					24.39; 6; <0.001				20.99; 6; 0.002
Below average	203 (4.7)	50.2 ^a,b^	36.0 ^a,b^	13.8 ^a,b^		36.0 ^a^	50.7 ^a^	13.3 ^a,b^	
Approx. average	2335 (54.4)	48.4 ^a,b^	41.9 ^b^	9.7 ^b^		38.9 ^a^	50.4 ^a^	10.6 ^b^	
Above average	1647 (38.3)	45.9 ^b^	40.9 ^b^	13.2 ^a^		37.8 ^a^	47.9 ^a^	14.3 ^a^	
Unknown	110 (2.6)	59.1 ^a^	26.4 ^a^	14.5 ^a,b^		51.8 ^b^	38.2 ^a^	10.0 ^a,b^	
Patients with little social support or limited informal care in the practice					102.00; 6; <0.001				54.60; 6; <0.001
Below average	937 (21.8)	55.0 ^a^	35.8 ^a^	9.3 ^a^		42.5 ^a^	47.6 ^a,b^	9.9 ^a^	
Approx. average	2336 (54.4)	47.3 ^b^	42.6 ^b^	10.0 ^a^		37.9 ^a^	50.2 ^b^	11.9 ^a^	
Above average	763 (17.8)	36.2 ^c^	45.2 ^b^	18.6 ^b^		31.7 ^b^	51.4 ^b^	16.9 ^b^	
Unknown	259 (6.0)	60.2 ^a^	30.1 ^a^	9.7 ^a^		52.5 ^c^	39.4 ^a^	8.1 ^a^	

^a,b,c^ The presented proportions per category of disclosure of and screening for DV with different superscripts differ significantly (post hoc χ^2^ test *p* > 0.05). Notes: Because the comparisons in this table involved 11 independent tests, we adopted a Bonferroni-corrected significance level of 0.050/11 = 0.004 for these analyses. Abbreviations: DV  =  Domestic Violence; GP = General Practitioner; Approx. = Approximately.

**Table 2 ijerph-20-03519-t002:** Results of the mixed-model ordinal logistic regressions (Model V) of the factors related to disclosure of DV by the patient to the GP and screening for DV by the GP during the COVID-19 pandemic.

	Disclosure of DV		Screening for DV	
	Exp(B) Odds Ratio	95% CI Odds Ratio	Exp(B) Odds Ratio	95% CI Odds Ratio
Number of GPs (ref. 1)				
2	1.09	1.03–1.16	-	-
3–4	1.13	1.07–1.19	-	-
5+	1.08	1.02–1.15	-	-
Unknown	1.07	0.91–1.26	-	-
Patients with psychiatric vulnerability (ref. Below average)				
Approx. average	1.13	1.07–1.19	1.09	1.04–1.15
Above average	1.28	1.20–1.36	1.13	1.06–1.20
Unknown	1.10	0.99–1.22	1.01	0.92–1.11
Screening for financial problems (ref. Not at all/less than before)				
As much as before	1.22	1.15–1.28	1.56	1.48–1.63
(Much) more than before	1.31	1.24–1.39	1.71	1.62–1.81
Unknown	1.42	1.11–1.82	1.53	1.21–1.94
Disclosure of financial problems (ref. Not at all/less than before)				
As much as before	1.22	1.14–1.30	1.14	1.07–1.21
(Much) more than before	1.30	1.22–1.39	1.18	1.11–1.25
Unknown	1.66	0.84–3.31	1.09	0.57–2.08
Patients with previous DV actively contacted (ref. No)				
Yes	1.35	1.27–1.43	1.29	1.22–1.37
Unknown	1.01	0.96–1.07	1.05	1.00–1.11
Triage info present in consultation room (ref. No)				
Yes, in print	-	-	1.10	1.05–1.15
Yes, electronically	-	-	1.12	1.00–1.25
Unknown	-	-	1.06	1.00–1.13
GP more involved in actively reaching out to patients (ref. Strongly disagree)				
Disagree	-	-	1.08	0.98–1.18
Neutral	-	-	1.02	0.94–1.12
Agree	-	-	1.10	1.01–1.20
Strongly agree	-	-	1.12	1.02–1.23
Unknown	-	-	1.01	0.90–1.13
Incidents (scale): Patients with urgent condition were seen late (ref. None)				
1 type	1.05	0.98–1.12	-	-
2 types	0.99	0.92–1.07	-	-
3 types	1.00	0.91–1.09	-	-
4 types	1.05	0.95–1.17	-	-
5 types	1.18	1.07–1.30	-	-
Unknown	1.02	0.97–1.08	-	-
Communication (scale) presence of website, leaflets, answering machine (None)				
1 item	-	-	1.02	0.96–1.08
2 items	-	-	1.06	0.95–1.18
3 items	-	-	1.21	1.07–1.38
4 items	-	-	1.15	0.99–1.35
Unknown	-	-	1.02	0.97–1.07

## Data Availability

The anonymized data is held at Ghent University and is available to participating partners for further analysis upon signing an appropriate usage agreement.

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
