# Peer review of "Screening for and Disclosure of Domestic Violence during the COVID-19 Pandemic: Results of the PRICOV-19 Cross-Sectional Study in 33 Countries"

_ijerph, 2023, doi:10.3390/ijerph20043519_

Round 1
Reviewer 1 Report
Thank you for inviting me to review this article. This article analyzes the prevalence of domestic violence during the COVID-19 pandemic, as well as related influencing factors. I have the following suggestions for authors to consider.
- In the introduction section, the authors reviewed the existing literature, but it lacked clear theoretical clues, and they did not construct a clear theoretical framework.
- In the method section, Cronbach's Alpha of (Pro-)active communication scale is lower than 0.7. I lack confidence in the effectiveness of this scale.
- In the results section, the authors can provide more goodness of fit criteria for statistical models, such as Pseudo-R2, BIC, etc.
- In the conclusion section, the authors may need to further deepen the policy implications of the findings.
Reviewer 2 Report
Thank you for giving me the opportunity to review this study, so I make the following recommendations.
Is the online questionnaire that was used validated? What are the questions in the questionnaire?
Why or for what purpose was a pilot study conducted?
Reviewer 3 Report
The authors described the frequency of screening for DV by GPs and disclosure of DV by patients to the GP during the COVID-19 pandemic and identified key elements that could potentially explain differences in screening for and disclosure of DV. Overall, the manuscript is well-written and I have only several minor feedback:
1) 1. Introduction: (a) Please provide the research hypotheses/questions as well as justifications. (b) For the 2nd aim, it would be better to discuss the literature on the potential key elements, so that the readers have a better idea of why this study is needed.
2) 2.3.3.(. Pro-)active communication: Please provide justification on why financial problems items are related to DV.
3) 2.4. Data Analysis: Kindly describe each statistical analysis according to research hypotheses/questions, so that the reader can easily know how each hypothesis is verified.
4) 4. Discussion: The "key elements that could potentially explain differences in screening for and disclosure of DV" is mentioned in both Abstract and Introduction. (a) Please elaborate on the meaning of differences in screening for and disclosure of DV. Is the differences examined in this study? (b) Is (pro)active communication the only "key element that could potentially explain differences in screening for and disclosure of DV" found in this study?
